# Complete Mitogenomic Structure and Phylogenetic Implications of the Genus *Ostrinia* (Lepidoptera: Crambidae)

**DOI:** 10.3390/insects11040232

**Published:** 2020-04-07

**Authors:** Nan Zhou, Yanling Dong, Pingping Qiao, Zhaofu Yang

**Affiliations:** 1Key Laboratory of Plant Protection Resources and Pest Management, Ministry of Education, Northwest A&F University, Yangling 712100, China; 2Entomological Museum, College of Plant Protection, Northwest A&F University, Yangling 712100, China

**Keywords:** mitochondrial genome, Crambidae, *Ostrinia*, phylogenetic analysis

## Abstract

To understand mitogenome characteristics and reveal phylogenetic relationships of the genus *Ostrinia,* including several notorious pests of great importance for crops, we sequenced the complete mitogenomes of four species: *Ostrinia furnacalis* (Guenée, 1854), *Ostrinia nubilalis* (Hübner, 1796), *Ostrinia scapulalis* (Walker, 1859) and *Ostrinia zealis* (Guenée, 1854). Results indicate that the four mitogenomes—*O. furnacalis*, *O. nubilalis*, *O. scapulalis,* and *O. zealis*—are 15,245, 15,248, 15,311, and 15,208 bp in size, respectively. All four mitogenomes are comprised of 37 encoded genes and a control region. All 13 protein-coding genes (PCGs) initiate with ATN and terminate with TAN, with the exception of *cox1* that starts with CGA, and *cox1*, *cox2*, and *nad5* that terminate with an incomplete codon T. All transfer RNA genes (tRNAs) present the typical clover-leaf secondary structure except for the *trnS1* (AGN) gene. There are some conserved structural elements in the control region. Our analyses indicate that *nad6* and *atp6* exhibit higher evolution rates compared to other PCGs. Phylogenetic analyses based on mitogenomes using both maximum likelihood (ML) and Bayesian inference (BI) methods revealed the relationship (*O. palustralis* + (*O. penitalis* + (*O. zealis* + (*O. furnacalis* + (*O. nubilalis* + *O. scapulalis*))))) within *Ostrinia*.

## 1. Introduction

The genus *Ostrinia* (Hübner) belongs to the subfamily Pyraustinae (Lepidoptera: Crambidae) [1,2,3] and includes several well-known agricultural pests (e.g., *O. furnacalis* and *O. nubilalis*) that cause huge losses of crops around the world [4,5,6]. Globally, *Tichogramma* spp. [7], transgenic crops [8], and sex pheromone traps [9] have been conducted as control strategies to reduce the threat to crops caused by *Ostrinia* spp. As different species exhibit distinct responses to specific biocontrol agents and pesticides, accurate species-level identification is very important in pest management. However, species identification of *Ostrinia* spp. is exceptionally difficult due to the many sibling species that are morphologically indistinguishable. Additionally, the phylogenetic relationships of this genus based on molecular data conflicts with the morphology-based taxonomy in previous studies. Mutuura and Munroe (1970) divided this genus into three species groups [2]. The first species group contains a single American species, *O. penitalis* (Grote, 1876), which has a trifid juxta and unarmed sacculus in the male genitalia. The second species group includes nine species with the sacculus dorsally spined and a simple or bifid uncus in the male genitalia. The third species group, also named the trifid-uncus species group, consists of ten species mainly occurring in Europe, East Asia, and Northwestern Africa. Within this third species group, three subgroups are recognized according to the differences in male mid-tibia. The first subgroup contains four species including *O. nubilalis* and *O. furnacalis* with smaller mid-tibia and lacking any groove or scales. The second subgroup includes two species, namely *O. kurentzovi* (Mutuura and Munroe, 1970) and *O. narynensis* (Mutuura and Munroe, 1970), with a moderately dilated tibia and a fringe of enlarged curved scales. The third subgroup consists of four species, including *O. scapulalis* and *O. zealis*, possessing a strongly dilated tibia and massive scales in the groove.

Mutuura and Munroe (1970) stated that the three species groups within the genus *Ostrinia* are monophyletic and indicated that *O. penitalis* is the most primitive species. The second group was defined as monophyletic because the members share potential synapomorphies in male genitalia [2]. The third species group has been intensively studied on the basis of their morphology, pheromones, and DNA sequence. Based on the complexity of the male tibia, Mutuura and Munroe (1970) considered that the species with the smallest tibia is the most primitive, then the species with the medium tibia are intermediate, and finally, the species with the large tibia and massive tuft are the most derived. 

Frolov (1981, 1984) stated that the variation in the male mid-tibia was determined by two alleles. The *Mt* and *Mt^+^* were two alleles located on autosomes and controlled the size of the mid-tibia, while *i* and *i^+^* were two alleles located on Z-sex-linked chromosomes and controlled the groove appearance. *Mt/Mt^+^* and *i/i^+^* follow a Mendelian inheritance pattern [10,11]. Additionally, Frolov (1981, 1984) found that the *O. scapulalis* (sensu lato) feeding on dicotyledons may be a single species with polymorphic male-tibia rather than three distinct species regardless of the size of the male-tibia [12,13,14]. Kim et al. (1999) inferred the phylogenetic relationship among second and third species groups by using the mitochondrial gene *COII* and found that two species groups were both monophyletic [15]. One cluster included two species with a bifid uncus (group II) and another comprised of six trifid-uncus species (group III). Interestingly, within this third species group, *O. scapulalis* (with a massive mid-tibia) was clustered into a clade with *O. nubilalis* (with small mid-tibia). Similarly, Yang et al. (2011) found that the relationship among four trilobed species was (*O. zealis* + (*O. furnacalis* + (*O. nubilalis* + *O. scapulalis*))) based on the mitochondrial gene *COI* [16]. Lassance et al. (2013) revealed the same topology of the genus based on both the mitochondrial genes and nuclear genes as in Kim et al. (1999) and Yang et al. (2011) [17]. These results imply that the relationship among species within the third group inferred through molecular data was incongruent with Mutuura and Munroe (1970). The phylogenetic relationships among members within this genus remain controversial. In particular, it is uncertain what contributes to delimiting the boundaries between species groups and taxa which are largely based on male genitalia. Furthermore, is the male tibia a polymorphism or stable character valid to subdivide members within the third species group?

The insect mitochondrial genome is a compact circular double-stranded molecule comprised of 37 encoding genes: 13 protein-encoding genes (PCGs), 22 transfer RNA genes (tRNAs), and two ribosomal RNA genes (rRNAs). It contains a special noncoding sequence called the control region or A+T-rich region [18]. The mitochondrial genome has been widely used to study taxonomy, population genetics, and phylogenetic relationships because of the extremely low rate of recombination, maternal inheritance, and faster evolutionary rate compared to nuclear DNA [19,20,21,22]. The mitogenome, in general, provides more phylogenetic information due to significant sequence differences that could be observed among related species compared to a single mitochondrial gene or a few genes [23,24,25,26,27]. 

However, mitogenome sequences of *Ostrinia* spp. are still lacking. Only one complete mitogenome (group II *O. palustralis*) and three partial mitogenomic sequences (group I *O. penitalis*, group III *O. nubilalis,* and *O. furnacalis*) have been reported in previous studies [28,29,30]. In this study, we attempt to sequence the complete mitogenomes of *O. furnacalis, O. nubilalis, O. scapulalis,* and *O. zealis* and reconstruct the phylogenetic relationships within the three species groups of the genus *Ostrinia*.

## 2. Materials and Methods 

### 2.1. Specimen Collection and DNA Isolation

All specimens for the four species, *O. furnacalis*, *O. nubilalis*, *O. scapulalis*, and *O. zealis* were collected by light trap or sweeping net from different localities in China and preserved either in a dried condition under 20 °C or in 100% ethanol in a −20 °C freezer (Appendix A). Samples were identified based on external morphological and male genital characters according to Mutuura and Munroe (1970) [2]. In order to confirm the identities, we generated the 658 base pair (bp) barcode region of *COI* sequences from a single leg using the DNAeasy DNA Extraction kit following the manufacturer’s protocols. LepF1/LepR1 primers were used to PCR amplify DNA fragments from the *COI* gene [31]. DNA products were subsequently bidirectionally sequenced by Sangon Biotechnology Co. Ltd (Shanghai, China). The *COI* sequences were cross-checked against the Barcode of Life Database [32]. Results showed that all samples were up to 99.8% similar to published *Ostrinia* species. For sequencing mitochondrial genome sequences, genomic DNA was extracted from thorax tissues by using a genomic DNA extraction kit following the manufacturer’s protocol (Biomarker, Beijing, China). All samples were deposited at the Entomological Museum, Northwest A&F University, Yangling, Shaanxi, China. Two additional mitochondrial genome sequences were downloaded from the GenBank for *O. penitalis* and *O. palustralis* [28,30].

### 2.2. Sequence Analysis and Gene Annotation

Complete mitogenomes were sequenced using the Illumina HiSeq platform with paired reads of 2 × 150 bp at Biomarker Technologies Co. Ltd (Beijing, China). MitoZ v 2.4 was implemented to construct annotated mitogenome from raw data under the *all* module by default [33]. The *all* module includes four steps: raw data pre-treatment, de novo assembly with SOAPdenvo-Trans, mitogenome sequence identification, and mitogenome annotation. All annotated mitogenomes were rechecked using Geneious v 11.0.2 (Biomatters, Auckland, New Zealand) based on previously published *Ostrinia* mitogenomes [34]. Correctly annotated mitogenomes were illustrated using the *visualize* module of MitoZ which employed Circos to show the gene element and sequence depth distribution. The tRNAs secondary structure was predicted using MITOS Web Serve (http://mitos2.bioinf.uni-leipzig.de/index.py) [35] with invertebrate mitochondrial gene codes then edited manually using Adobe Illustrator CC2019 according to the predicted results.

Nucleotide composition, composition skew, codon usage, relative synonymous codon using (RSCU), and architecture tables were calculated by PhyloSuite v 1.2.1 [36]. The tandem repeats of the A+T-rich region were established by Tandem Repeats Finder Online server (http://tandem.bu.edu/trf/trf.html) [37]. The sliding window analysis (a sliding window of 200 bp and a step size of 20) and evolutionary rate analysis were conducted with DnaSP v 5.0 based on 13 aligned protein-coding genes (PCGs) [38]. Genetic distances between species based on each PCG was estimated using Mega v 7.0 with Kimura-2-parameter [39]. In this study, the four newly-sequenced mitogenomes were deposited in GenBank with the accession numbers of MN793322-MN793325.

### 2.3. Phylogenetic Analysis 

For phylogenetic analysis (Table 1), both *O. penitalis* and *O. palustralis* were added as ingroups. Two published species mitogenomes, *Loxostege sticticalis* (Crambidae: Pyraustinae, Linnaeus, 1761) and *Cnaphalocrocis medinalis* (Crambidae: Spilomelinae, Guenée, 1854), were selected as outgroups [28,30,40,41]. PhyloSuite v 1.2.1 was used to extract the mitochondrial genes. Then each PCG was aligned by MAFFT v 7.313 plugin in PhyloSuite using codon alignment mode [42]. Two rRNAs were aligned individually by the MAFFT-with-extensions software with the Q-INS-i strategy [42]. Gblocks v 0.91b was used to remove gaps and ambiguous sites [43]. Aligned PCGs and rRNAs were concatenated by PhyloSuite respectively [36].

Based on previous studies, 13PCGs and two rRNAs were widely used to construct a phylogenetic relationship. Moreover, replicate analyses including or excluding the third codons could test variability in the phylogenetic performance [20,44]. Herein, the phylogenetic analyses were reconstructed by the Bayesian inference (BI) and the maximum likelihood (ML) methods based on four datasets: all codon positions for the 13 PCGs (PCG123), all codon positions for the 13 PCGs and two rRNA genes (PCG123R), the 13 PCGs excluding the third codon position (PCG12), and the 13 PCGs excluding the third codon position and two rRNA genes (PCG12R). The best partitioning schemes and evolution models of both BI analyses and ML analyses were estimated using PartitionFinder v 2.1.1 integrated into PhyloSuite using the greedy search algorithm with branch lengths linked and Bayesian information criterion (BIC) [45]. The best-fit substitution results are shown in (Appendix A). BI analyses were performed using MrBayes v 3.2.6 with default settings and 5 × 10^6^ Markov chain Monte Carlo (MCMC) generations, sampled every 1000 generations [46]. The average standard deviation of split frequencies <0.01 was considered to reach convergence. The initial 25% of sampled data were discarded as burn-in. ML analyses were performed by using IQ-TREE v 1.6.8 under ultrafast bootstraps with 1000 replicates [47].

## 3. Results and Discussion

### 3.1. Mitogenome Structure and Organization

Five complete mitogenomes including the newly sequenced four mitogenomes in this study and the *O. palustralis* mitogenome downloaded from GenBank were analyzed. The length of entire mitogenome sequences ranged from 15,208 bp to 15,311 bp and included 37 genes (13PCGs, 22 tRNAs, and two rRNAs) and a control region. Among the 37 mitochondrial genes, 23 genes (9 PCGs and 14 tRNAs) were found on the majority strand (J-strand) and the remaining genes (4 PCGs, 8 tRNAs, and 2 rRNAs) were located on the minority strand (N-strand). Five mitogenomes show the typical circular double-stranded molecule structure and the same arrangement. The gene order is identical to those of mitogenomes in Pyraloidea and is characterized by *trnM-trnI-trnQ* while it differs from ancestral groups with *trnI-trnQ-trnM* [48,49,50] (Figure 1 and Appendix A).

All five complete mitogenomes show similar nucleotide composition. On average, five entire mitogenomes contain A = 41.74%, T = 39.14%, C = 11.50%, and G = 7.70% with a high A+T content congruent with those of other Pyraloidea [51]. The control region has the highest AT bias (94.24%) among the whole mitogenome. In general, all mitogenomes have a slightly positive AT-skew = 0.032 and negative GC-skew = −0.196 (Table 2).

In *O. furnacalis*, *O. nubilalis*, *O. scapulalis*, and *O. zealis* mitogenomes, 14 intergenic spacers (ranging from 1 bp to 62 bp) were observed with a total length from 170 bp to 175 bp. By contrast, there were 18 intergenic spacers in *O. palustralis* (ranging from 1 bp to 61 bp). All five mitogenomes show some similar intergenic spacers. Particularly, two intergenic spacers are common with most mitogenomes of Pyraloidea [52], including the longest (about 61 bp) intergenic spacer located between *trnQ* and *nad2* genes and the intergenic spacer (about 30 bp) located between *trnS2* and *nad1*. There were nine overlapping genes regions ranging from 1 bp to 8 bp in length in four mitogenomes (*O. furnacalis*, *O. nubilalis*, *O. scapulalis*, and *O. zealis*), while seven overlapping genes regions (ranging from 1 bp to 8 bp) were present in *O. palustralis*. Among the five mitogenomes, the longest overlapping sequence was located between *trnW* and *trnC* (about 8 bp) and the second was located between *atp8* and *atp6* (about 7 bp).

### 3.2. Protein-coding Genes

All PCGs among the five complete mitogenomes are similar in general character with a total length of about 11,163 bp (Appendix A). Four PCGs (*nad1*, *nad4*, *nad4L*, and *nad5*) were encoded on the N-strand; the others (*cox1*, *cox2*, *cox3*, *cytb*, *atp6*, *atp8*, *nad2*, *nad3*, and *nad6*) were encoded on the J-strand. Most PCGs initiate with ATN with the exception of the *cox1* that initiates with CGA. The majority of PCGs for these five complete mitogenomes terminated with TAA or TAG except for three genes (*cox1*, *cox2,* and *nad5*) that stopped with an incomplete codon T. By contrast, all PCGs in *O. penitalis* were represented by traditional TAG and TAA stop codons [30]. This incomplete terminating codon among arthropod mitogenomes is a common phenomenon, which might be related to post-transcriptional modification during mRNA maturation [53]. The relative synonymous codon usage (RSCU) is shown in (Figure 2). The codon usage of these six species is roughly the same. Overall codon usage analysis indicated that the codon ending up with T or A is more frequent than C or G. The most prevalent usage codons were UUA-Leu2, AUU-Ile, UUU-Phe, and AUA-Met, all composed with A and T which contribute to the high A + T bias of the entire mitogenomes. In total, the codon usage of these six *Ostrinia* spp. is very similar to those of other Pyraloidea [54].

### 3.3. Ribosomal and Transfer RNA Genes

For five complete mitogenomes, the length of *rrnL* ranges from 1332 bp to 1341 bp and is located between *trnL1* and *trnV*. The length of *rrnS* ranges from 778 bp to 779 bp and is located between *trnV* and the control region. Both *rrnL* and *rrnS* show a negative AT-skew and positive GC-skew and are encoded on the N-strand. The A+T content was roughly the same (ca. 85%) for the two rRNA genes among these five mitogenomes (Table 2).

The transfer RNA genes of five complete mitogenomes are dispersed among the genes of rRNA and PCG. The length of tRNA gene sequences ranges from 65 to 71 bp, and the entire tRNAs are 1480 bp on average. The tRNAs show both positive AT-skew and GC-skew. Twenty one tRNAs were observed to fold into a cloverleaf structure, except for the *trnS1* (AGN) in which the DHU arm forms a simple loop (Figure 3). The partial amino acid acceptor stems and anticodon loops are highly conserved, while the DHU and TΨC arms are variable. There were two frequent types of miss pairings including non-canonical G-U pairs and mismatched base pairs U-U. Among the five complete mitogenomes, *O. scapulalis* and *O. nubilalis* are characterized as the same transfer RNA genes and the four species with trilobed uncus (group III) are similar to each other. Of all tRNAs, the *trnG*, *trnS1*(AGN), and *trnT* are identical among species, implying the lowest variation. In contrast, the *trnI* and *trnH* indicate relatively higher nucleotide substitutions. These results indicate that the tRNAs are relatively conserved which is similar to other Pyraloidea [55].

### 3.4. Control Region

The control region, also called the A + T-rich region in insects [56], is the major non-coding region, which plays an important role in the study of molecular evolution [57]. This region contains regulatory elements for replication and transcription. Among five complete mitogenomes, this control region is flanked by *rrnS* and *trnM* with a length of 330 bp in *O. furnacalis*, 332 bp in *O. nubilalis*, 402 bp in *O. scapulalis*, 300 bp in *O. zealis*, and 330 bp in *O. palustralis*. This indicates that *O. scapulalis* possesses a much longer control region whereas the remaining four species are similar to other Pyraloidea (about 339 bp) [58,59].

The nucleotide composition of the control region among these five species exhibits both negative AT-skew and GC-skew, with A + T content approximately 94% (Table 2). The results indicate that the control region exhibits a high similarity of conserved structural elements among five species (Figure 4). Many conserved blocks reported in other published Pyraloidea mitogenomes are also found in the control region elements of these five mitogenomes [60]. We infer that three conserved blocks in the control region—e.g., a conserved block (about 25 bp) terminate with the motif ‘TTAGA’ preceded a long poly-T stretch, a varied and typical microsatellite-like (TA)n element, and a conserved block (about 34 bp)—end up with an ‘A-rich’ upstream of *trnM*. These conserved blocks may be involved in controlling the replication and transcription of the mitogenome. The motif ‘TTAGA + poly-T stretch’ detected in the genus *Ostrinia* differs from many sequenced Lepidoptera with ‘ATAGA + ploy-T stretch’ motif that resides downstream of *rrnS* [61], suggesting that the motif ‘TTAGA’ followed by a long poly-T stretch might be the origin of light-strand replication within the genus *Ostrinia*.

### 3.5. Nucleotide Diversity Analyses

In this study, the sliding window analysis was implemented to study the nucleotide diversity based on 13 aligned PCGs of *O. furnacalis*, *O. nubilalis*, *O. scapulalis*, *O. palustralis*, *O. zealis*, and *O. penitalis* (Figure 5A). Our results indicate that the *nad6* (pi = 0.055) and *atp6* (pi = 0.052) were found to have slightly higher nucleotide diversity than other genes, while the genes *cox2*, *cox3*, *nad2*, *nad3*, and *nad5* exhibit a relatively low nucleotide diversity of 0.040, 0.040, 0.036, 0.033, 0.034, and *atp8* shows the lowest value (pi = 0.018). Congruent results were observed according to pairwise genetic distance analysis: *atp6* (0.055) and *nad6* (0.058) are evolving comparatively faster, while *atp8* (0.019) and *nad3* (0.034) are evolving comparatively slow (Figure 5B). The evolutionary rate analysis estimated by the average non-synonymous (Ka) and synonymous (Ks) substitution rates of 13 PCGs among the six mitogenomes ranged from 0.009 to 0.614 (0 < ω < 1), indicating that all PCGs are under purifying selection. Additionally, *cox1* exhibits the strongest purifying selection with the lowest Ka/Ks value (0.009). Both *nad6* (0.083) and *atp6* (0.102) show slightly higher Ka/Ks values than most of other PCGs, indicating they are likely to be under a relaxed purifying selection. The *atp8* with the extremely higher (Ka/Ks = 0.614) values may be an outlier, similar results were observed in the previous study [62].

The mitochondrial gene *cox1* has been one of the commonly used markers and a powerful tool for identifying species and inferring the phylogenetic relationship due to it is easy amplification. However, the efficiency of species identification, particularly delimitation of closely-related taxa based on the *cox1* gene has received more concern [63]. In this study, we found that the *cox1* gene exhibits a relatively conserved and slow evolution rate compared to other PCGs, while *atp6* and *nad6* genes have a relatively faster evolution rate and evolve under comparative relaxed purifying selection, suggesting that *atp6* and *nad6* would be two suitable candidate markers for clarifying the phylogenetic relationships for indistinguishable sibling species. 

### 3.6. Phylogenetic Relationships

The phylogenetic relationships among six *Ostrinia* spp. were inferred based on four datasets (PCG123, PCG123R, PCG12, PCG12R). Our results indicate that the topology of ML and BI analyses was highly congruent (Figure 6), suggesting that the relationships within this genus are (*O. palustralis* + (*O. penitalis* + (*O. zealis* + (*O. furnacalis* + (*O. nubilalis* + *O. scapulalis*))))) with high support values. Both ML and BI trees revealed that the third species group is monophyletic. However, the relationships among them are different from the Mutuura and Munroe (1970) classification based on morphological data [2]. 

The representative of the first species group *O. penitalis* might not be the most primitive species as it may not constitute the basal elements of the genus instead of *O. palustralis*, which is the main difference to Mutuura and Munroe’s (1970) classification. Our results show that all members of the third group (*O. zealis*, *O. furnacalis*, *O. nubilalis*, and *O. scapulalis*) are grouped into a clade with high support values (PP = 1, BS = 100). Within this group, *O. furnacalis*, *O. nubilalis*, and *O. scapulalis* were clustered together and sister to *O. zealis*. We also found that *O. nubilalis* and *O. scapulalis* were placed into a subclade with strong nodal support values (PP = 1, BS = 100). Our findings highly support the phylogenetic relationship within the third group revealed by mitochondrial *COI* and *COII* in previous studies [15,16]. In other words, our results confirmed that the division of three subgroups within the third species group based on male mid-tibia by Mutuura and Munroe (1970) needs to be reconsidered. Frolov (1981, 1984) and Frolov et al. (2007, 2011) stated that the male mid-tibia structure was unstable and overstated in distinguishing the species of the genus *Ostrinia* [10,11,12,13,14]. For example, the mid-tibia differences may be just intraspecific polymorphisms of *O. scapulalis* and not reliable taxonomic characters to distinguish species among *Ostrinia*. Therefore, the mid-tibia polymorphism most likely misled Mutuura and Munroe (1970) to split the trifid-uncus group into three subgroups without any molecular evidence. Therefore, additional sampling of taxa and markers would be helpful to clarify this contradiction between the morphology-based and molecular-based phylogenies of the genus *Ostrinia*. 

## 4. Conclusions

In this study, we newly sequenced four complete mitogenomes including *O. furnacalis*, *O. nubilalis*, *O. scapulalis*, and *O. zealis*. Compared to other previously reported complete mitogenomes of *Ostrinia*, all of them have similar structural characters and nucleotide composition. All mitogenomes were composed of 37 typically encoded genes and a control region. Most PCGs initiate with ATN and terminate with TAN, while *cox1* starts with CGA and *cox1*, *cox2,* and *nad5* terminate with an incomplete codon T. All tRNAs can fold into a clover-leaf secondary structure except the *trnS1*(AGN) gene which lacks the DHU arm. Both the *rrnS* gene and *rrnL* gene are relatively conserved. Some conserved blocks in the control region are observed which may be typical structural elements among *Ostrinia* mitogenomes. It is worth mentioning that *nad6* and *atp6* show relatively higher variability and evolve under slightly relaxed purifying selection than other PCGs. The phylogenetic analysis indicates that (*O. palustralis* + (*O. penitalis* + (*O. zealis* + (*O. furnacalis* + (*O. nubilalis* + *O. scapulalis*) within *Ostrinia*. Furthermore, we proposed that *O. penitalis* might not be the most primitive species. The division of the third species group based on male mid-tibia needs to be reconsidered. 

## Figures and Tables

**Figure 1 insects-11-00232-f001:**
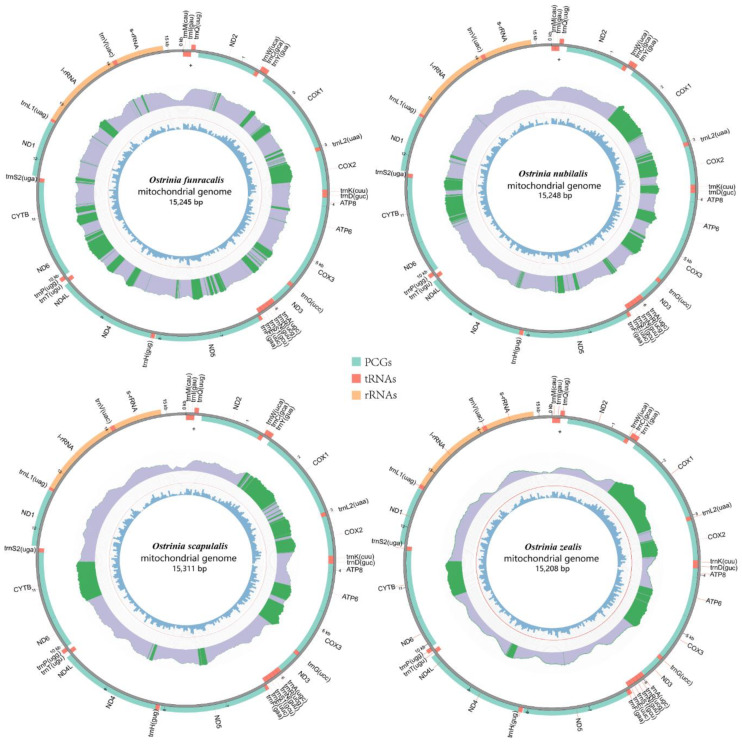
Complete mitochondrial genomes of four species. The innermost circle and middle circle indicate the GC content and read depth distribution, respectively. Light purple blocks show the depth upper than the minimum (default 20); the green blocks show the depth greater than the upper quartile and the green line indicates the outline. The outermost circle shows the arrangement of the genes: blue for the CDS, red for tRNAs, and orange for rRNAs.

**Figure 2 insects-11-00232-f002:**
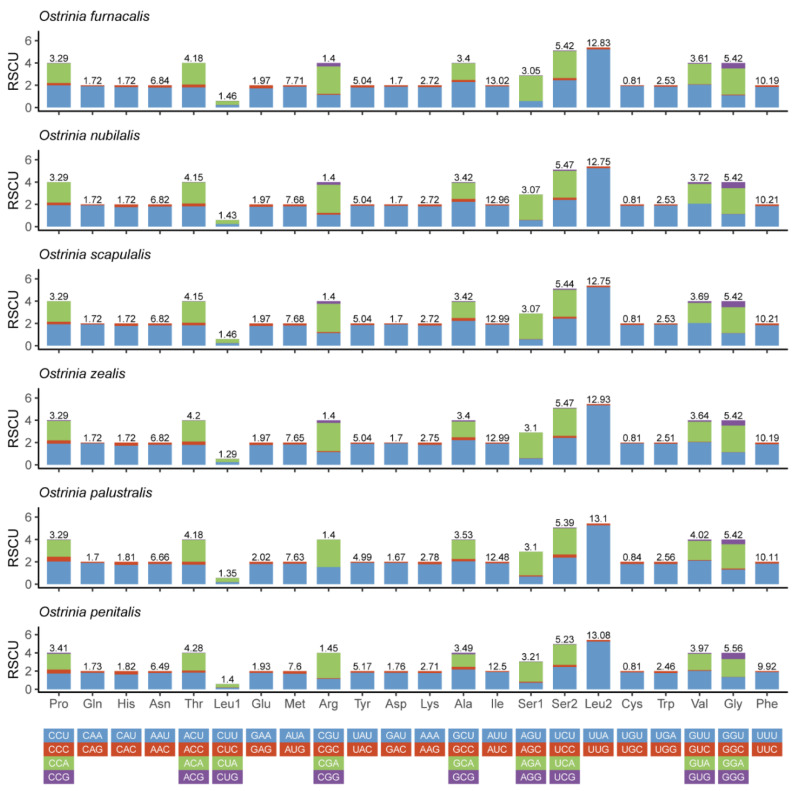
Relative synonymous codon usage (RSCU) in the mitogenomes of *Ostrinia* spp.

**Figure 3 insects-11-00232-f003:**
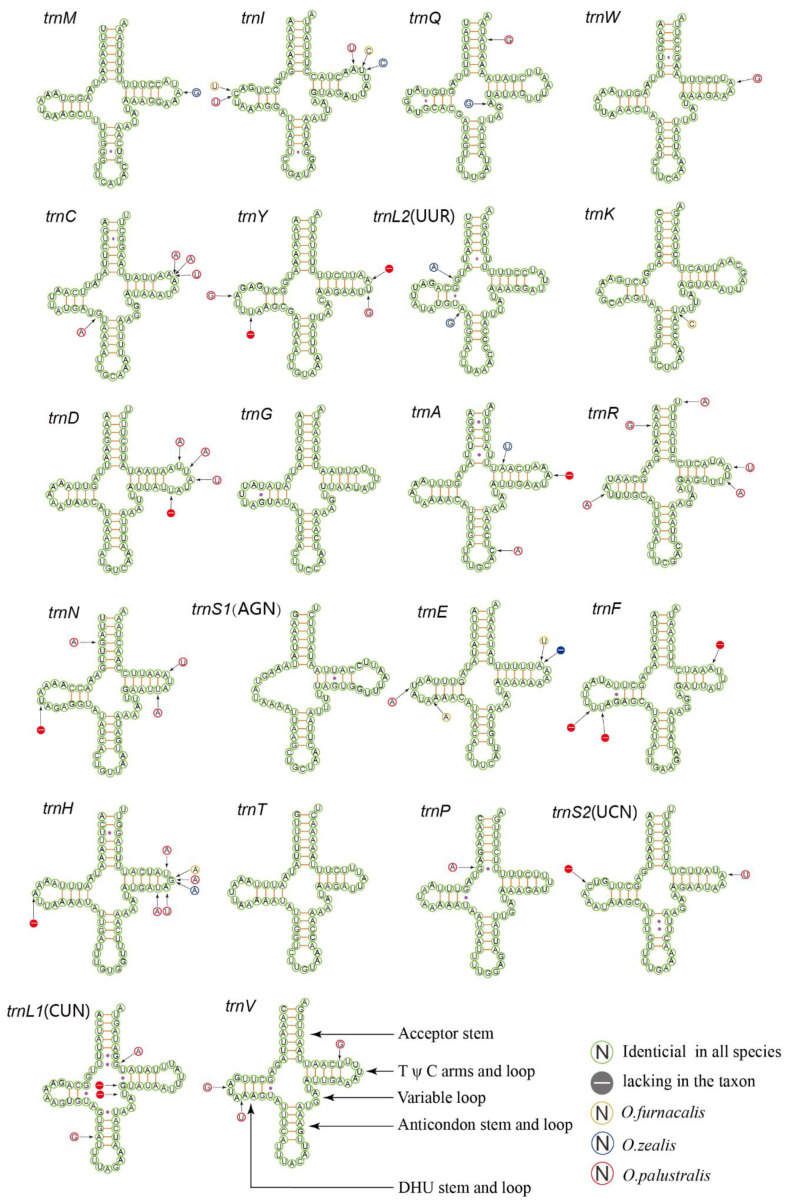
Predicted secondary cloverleaf structure for the tRNAs of *O. furnacalis*, *O. nubilalis*, *O. scapulalis*, *O. zealis*, and *O. palustralis*.

**Figure 4 insects-11-00232-f004:**
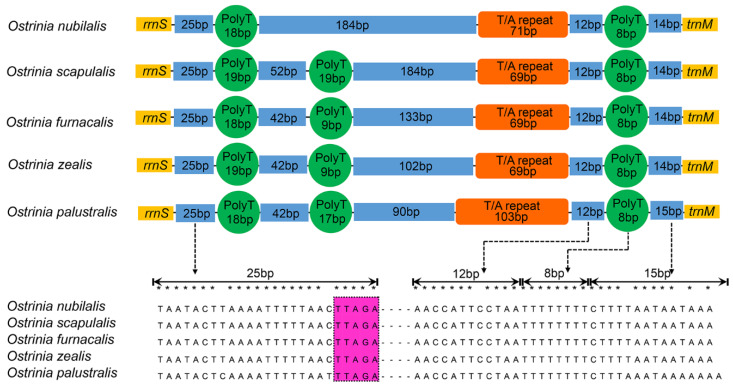
Organization of control region in mitogenomes of *O. furnacalis*, *O. nubilalis*, *O. scapulalis*, *O. zealis*, and *O. palustralis*. The orange block is the tandem repeat region, the dark blue block indicates non-repeat regions and the green circle is the poly-T stretch. The motif ‘TTAGA’ is highlighted by purple.

**Figure 5 insects-11-00232-f005:**
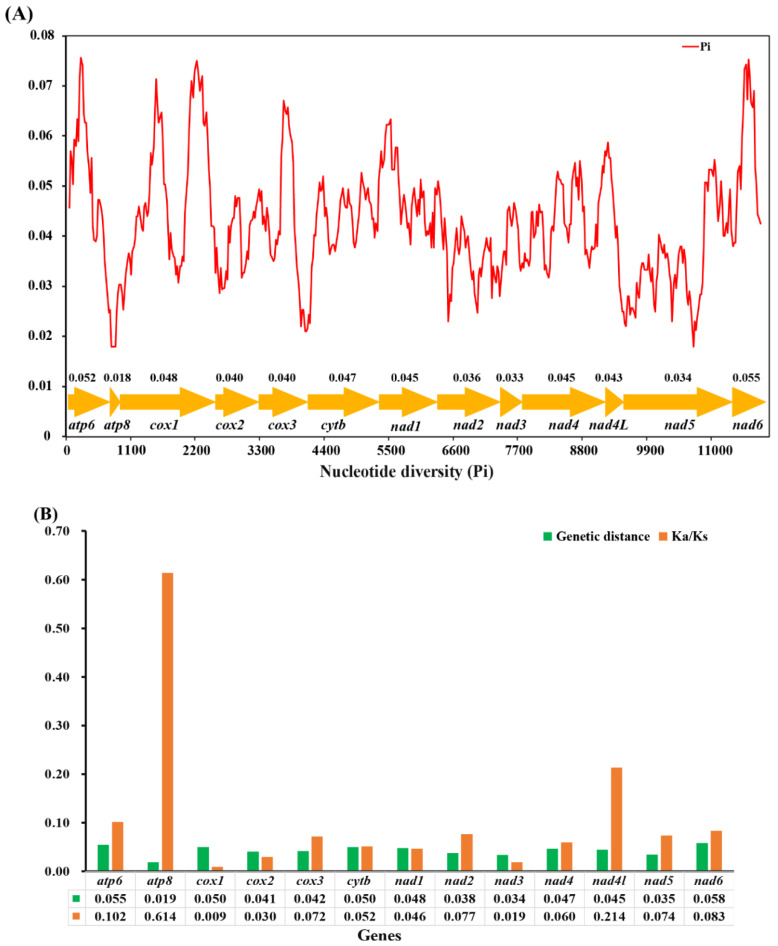
(**A**) Sliding window analysis based on 13 aligned PCGs. The red line shows the value of nucleotide diversity Pi (window size = 200 bp, step size =20 bp). The gene names and Pi values are shown in the graph. (**B**) Genetic distance (on average) and non-synonymous (Ka) to synonymous (Ks) substitution rates of 13 PCGs among six *Ostrinia* spp.

**Figure 6 insects-11-00232-f006:**
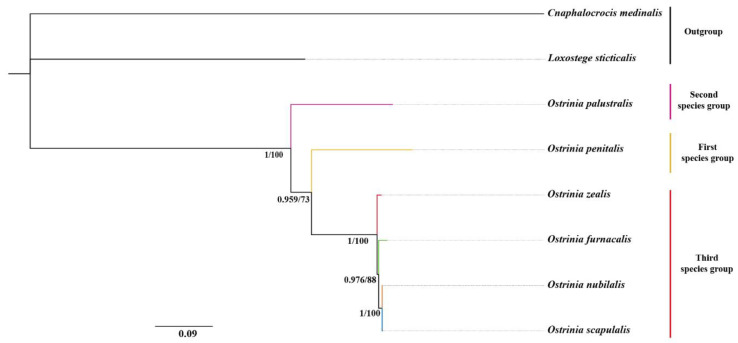
The phylogenetic tree of *Ostrinia* spp. based on the dataset of PCG123. ML and BI analyses show the same topology. The numbers under the branches are Bayesian posterior probabilities (PP) and bootstrap support values (BS). Scale bar represents nucleotide substitutions per site.

**Table 1 insects-11-00232-t001:** The mitogenomic sequences used in this study.

Subfamily	Species	Whole Length	GenBank Accession No.	Reference
Pyraustinae	*Loxostege sticticalis*	15,218	KR080490	[41]
	*Ostrinia palustralis*	15,246	MH574940	[28]
	*Ostrinia penitalis*	12,612	KM395814.1	[30]
	*Ostrinia furnacalis*	15,245	MN793323	This study
	*Ostrinia nubilalis*	15,248	MN793322	This study
	*Ostrinia scapulalis*	15,311	MN793324	This study
	*Ostrinia zealis*	15,208	MN793325	This study
Spilomelinae	*Cnaphalocrocis medinalis*	15,368	NC_022669	[40]

**Table 2 insects-11-00232-t002:** Nucleotide composition and skewness of five *Ostrinia* spp. mitogenomes.

Regions	Species	Size (bp)	T%	C%	A%	G%	AT(%)	GC(%)	AT Skew	GC Skew
Full genome	*O. furnacalis*	15,245	39.2	11.4	41.8	7.6	81.0	19.0	0.032	−0.199
	*O. nubilalis*	15,248	39.2	11.5	41.7	7.7	80.9	19.2	0.031	−0.196
	*O. scapulalis*	15,311	39.3	11.4	41.7	7.7	81.0	19.1	0.030	−0.196
	*O. zealis*	15,208	39.2	11.4	41.7	7.7	80.9	19.1	0.031	−0.193
	*O. palustralis*	15,246	38.8	11.6	41.8	7.8	80.6	19.4	0.036	−0.198
PCGs	*O. furnacalis*	11,163	45.0	9.9	34.5	10.6	79.5	20.5	−0.132	0.036
	*O. nubilalis*	11,163	45.0	9.9	34.4	10.7	79.4	20.6	−0.133	0.042
	*O. scapulalis*	11,163	45.0	9.8	34.4	10.7	79.4	20.5	−0.133	0.043
	*O. zealis*	11,163	45.0	9.8	34.5	10.7	79.5	20.5	−0.132	0.040
	*O. palustralis*	11,160	44.9	10.0	34.2	10.8	79.1	20.8	−0.135	0.039
1st codon position	*O. furnacalis*	3721	37.0	9.6	37.4	16.0	74.4	25.6	0.006	0.253
	*O. nubilalis*	3721	37.0	9.5	37.3	16.2	74.3	25.7	0.004	0.258
	*O. scapulalis*	3721	37.0	9.6	37.3	16.2	74.3	25.8	0.005	0.256
	*O. zealis*	3721	37.1	9.4	37.4	16.1	74.5	25.5	0.004	0.262
	*O. palustralis*	3720	37.2	9.5	36.7	16.6	73.9	26.1	−0.006	0.272
2nd codon position	*O. furnacalis*	3721	48.7	16.2	21.9	13.2	70.6	29.4	−0.379	−0.104
	*O. nubilalis*	3721	48.6	16.3	21.9	13.2	70.5	29.5	−0.379	−0.105
	*O. scapulalis*	3721	48.6	16.3	21.9	13.2	70.5	29.5	−0.379	−0.104
	*O. zealis*	3721	48.6	16.3	21.9	13.2	70.5	29.5	−0.378	−0.106
	*O. palustralis*	3720	48.5	16.3	21.8	13.3	70.3	29.6	−0.380	−0.103
3rd codon position	*O. furnacalis*	3721	49.3	3.8	44.2	2.6	93.5	6.4	−0.055	−0.188
	*O. nubilalis*	3721	49.3	3.8	44.0	2.9	93.3	6.7	−0.056	−0.141
	*O. scapulalis*	3721	49.4	3.7	44.1	2.8	93.5	6.5	−0.057	−0.132
	*O. zealis*	3721	49.4	3.8	44.2	2.7	93.6	6.5	−0.055	−0.167
	*O. palustralis*	3720	49.1	4.2	44.1	2.6	93.2	6.8	−0.054	−0.234
tRNAs	*O. furnacalis*	1480	39.8	7.8	41.7	10.7	81.5	18.5	0.023	0.161
	*O. nubilalis*	1480	39.8	7.7	41.7	10.8	81.5	18.5	0.023	0.168
	*O. scapulalis*	1479	39.8	7.7	41.6	10.8	81.4	18.5	0.022	0.168
	*O. zealis*	1477	39.9	7.8	41.4	10.9	81.3	18.7	0.019	0.167
	*O. palustralis*	1481	39.7	7.3	42.1	10.9	81.8	18.2	0.030	0.197
rRNAs	*O. furnacalis*	2120	43.3	4.8	42.0	9.9	85.3	14.7	−0.016	0.350
	*O. nubilalis*	2118	43.3	4.8	42.0	9.9	85.3	14.7	−0.015	0.350
	*O. scapulalis*	2117	43.3	4.8	42.0	9.9	85.3	14.7	−0.015	0.348
	*O. zealis*	2117	43.3	4.8	42.0	9.9	85.3	14.7	−0.014	0.344
	*O. palustralis*	2110	43.5	4.9	41.2	10.4	84.7	15.3	−0.027	0.362
Control region	*O. furnacalis*	330	51.2	4.5	42.7	1.5	93.9	6.0	−0.090	−0.500
	*O. nubilalis*	332	51.2	4.5	42.5	1.8	93.7	6.3	−0.093	−0.429
	*O. scapulalis*	402	52.5	4.5	41.3	1.7	93.8	6.2	−0.119	−0.440
	*O. zealis*	300	51.7	5.0	42.3	1.0	94.0	6.0	−0.099	−0.667
	*O. palustralis*	330	50.3	3.3	45.5	0.9	95.8	4.2	−0.051	−0.571

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
