# Peer review of "Complete Mitogenomic Structure and Phylogenetic Implications of the Genus Ostrinia (Lepidoptera: Crambidae)"

_insects, 2020, doi:10.3390/insects11040232_

Round 1
Reviewer 1 Report
Review of a manuscript titled “Complete mitogenomic structure and phylogenetic implications of the genus Ostrinia (Lepidoptera: Crambidae)”
This manuscript was a concise and clear representation -- with nice illustrations -- of a body or work revealing the complete mitogenomes of four moth species in the genus Ostrinia, some of which are notorious pests of important crop species (e.g. corn). I was not familiar with all the methods used for describing and illustrating mitogenomes, but considered the bioinformatics and results sufficiently and clearly described. I have no major concerns regarding the soundness of the work. However, given the sampling of both taxa (only four species from two subgroups) and molecular resources (limited to mitochondria only), I found the current framing of the work somewhat unsuited.
The authors justify their efforts by providing more resolution to the species -level phylogeny for these moths, many of which lack clear morphological apomorphies (traits unique to particular species), and also by providing molecular resources for various further studies on the selected species. To avoid potential pitfalls detailed below, I suggest rephrasing the study aims, and reframing introduction and discussion more towards the latter of the two incentives above. Although phylogenetic relationships within Ostrinia are interesting, I see more value in the molecular resources provided by this work than in the information value added to the already existing phylogenetic hypothesis.
Main concerns
1) The purpose of the study (from wide, societal point of view)
At the beginning of the introduction (lines 29-34) the authors (referred to as “you” from now on) state that this group of species (Ostrinia) includes pests, are hard to identify and have a spurious phylogeny. However, I am failing to see the link between species identification and pest management – how are these species controlled? If a general pesticide is used, why is species identification necessary? If the species have distinct pheromones that are used in pest management, then please make this connection clear in the introduction. Furthermore, if knowing the full mitogenome is helpful for the management in some way, please explain how.
2) The purpose of the study (from narrow, phylogenetic point of view)
Study aims (on lines 82-84) are stated as follows: “…to sequence the complete mitogenomes of O. furnacalis, O. nubilalis, O. scapulalis, and O. zealis and reconstruct the phylogeny within the three species groups of the genus Ostrinia.” Please clarify, whether you refer to the three species groups as delineated by Mutuura and Monroe (1970), or to the three subgroups within Mutuura and Monroe’s group III. Based on sampled taxa, the aim was to identify species relationships among the latter smaller group, as only four species all belonging to this subgroup were sampled.
3) Not enough evidence to “debate” the existing phylogenetic hypothesis?
According to your introduction, Kim et al. (1999) and Yang et al. (2011) had already resolved the phylogeny of the Ostrinia species considered here, using also mitochondrial genes – see lines 67-69 and your results. You also state on line 287-289: “Our findings highly support the phylogenetic relationship within the third group revealed by mitochondrial COI and COII in previous studies [11,52]”. Therefore it seems, that you are not really “debating” the current hypothesis of the real phylogeny, but rather just providing more detailed picture of what can be inferred using mitochondrial data only. Even though you have higher support for species relationships using whole mitogenomes than Kim et al. (1999) and Yang et al. (2011) using either COII or COI, your results are essentially the same. This makes me question, why did you decide to test the phylogenetic hypothesis using mitochondrial markers only? Did you expect to find different results based on more mitochondrial genes or regions than with COII or COI alone? To me, this does not constitute as a robust test for the species relationships (see my points number 4-6 below).
If you want to make the novelty of your results clear, please clarify what is different between the previous results and your results. However, if the motivation of this study was rather to solve the mitochondrial genomes fully for some further use than to “debate the phylogenetic relationships of the genus Ostrinia”, as you state in the very first sentence of your abstract (line 12), please change the abstract and introduction accordingly. The last sentence of the introduction (lines 85-87) describes your motivation nicely, but could be expanded on. I.e. I am not fully convinced that resolving the mitogenome will help facilitate studies on “e.g. morphological character adaptations, host preferences, biogeographical origins or pheromones”, unless you provide some examples of how this can be done (i.e. references for examples of each of the suggested applications, or explanation on how this is relevant). Also, in the last paragraph of your discussion, you could tone down the sentences debating the phylogeny (i.e. line 289-290), where you propose that the division based on male mid-tibia needs to be reconsidered – this is already evident from previous studies.
4) Incongruent mitochondrial vs. nuclear gene trees
If the main aim of this study was to resolve the true phylogenetic relationships, why did you choose to use the same genetic data source already used, and not e.g. nuclear genes or more variable microsatellites? Trees based on mitochondrial genes have often been found to deviate from trees inferred from nuclear genes, see e.g. Funk and Omland 2003, Leache and McGuire 2006. Mitochondria may be introgressed between closely related species if they interbreed, which would result in a different phylogeny inferred from mitochondrial genes than from nuclear genes. Have you checked the possibility for interbreeding in Ostrinia?
Funk, D. J. & Omland, K. E. Species-level paraphyly and polyphyly: frequency, causes, and consequences, with insights from animal mitochondrial DNA. Annu. Rev. Ecol. Evol. Syst. 34, 397–423 (2003).
Leache, A. D. & McGuire, J. A. Phylogenetic relationships of horned lizards (Phrynosoma) based on nuclear and mitochondrial data: evidence for a misleading mitochondrial gene tree. Mol. Phylogenet. Evol. 39, 628–644 (2006).
5) Incomplete lineage sorting
If introgression occurs only in some populations but not others, it may result in incomplete lineage sorting. Now if you sampled only from one population, and one individual of each species, how can you account for possible incomplete lineage sorting? This effect has been shown to be a likely cause for incongruent species trees between mtDNA and nuclear DNA (e.g. Wang et al. 2018).
Wang, K., Lenstra, J. A., Liu, L., Hu, Q., Ma, T., Qiu, Q., & Liu, J. (2018). Incomplete lineage sorting rather than hybridization explains the inconsistent phylogeny of the wisent. Communications biology, 1(1), 1-9.
6) Limited sampling effort
If your aim was to resolve species relationships within the three subgroups in the third species group in Ostrinia, then this sampling is fine. However, if you aimed for the three species groups, why did you not sample more species? If you aimed to provide resources for further studies on e.g. pheromones, why not sequence genes associated with pheromones or even whole genomes for these species? If the reason was cost-efficiency, please state so.
7) Wolbachia infection
Did you check whether any of you samples were infected with Wolbachia, or detect any traces of a possible Wolbachia infection, which may have caused changes in the mitogenome of any of the species? Wolbachia infection may cause a selective sweep in the genome reducing sequence diversity, but also increased levels of DNA introgression (perhaps also between species). For examples of genomic effects of Wolbachia in Lepidoptera, see e.g. Jiggins 2003, Narita et al. 2006, Narita et al. 2007 and for introgression also Raychoudhury et al. 2009 on wasps. Please elaborate whether you checked for Wolbachia infection, and whether you find it necessary to consider the possibility and importance of Wolbachia infection to the structure of these four new mitogenomes and the inferred phylogeny?
Jiggins, F. M. 2003. Male‐killing Wolbachia and mitochondrial DNA: selective sweeps, hybrid introgression and parasite population dynamics. Genetics 164:5–12.
Narita, S., M. Nomura, Y. Kato, and T. Fukatsu. 2006. Genetic structure of sibling butterfly species affected by Wolbachia infection sweep: evolutionary and biogeographical implications. Mol. Ecol. 15:1095–1108.
Narita, S., M. Nomura, Y. Kato, O. Yata, and D. Kageyama. 2007. Molecular phylogeography of two sibling species of Eurema butterflies. Genetica 131:241–253.
Raychoudhury, R., L. Baldo, D. C. S. G. Oliveira, and J. H. Werren. 2009. Modes of acquisition of Wolbachia: horizontal transfer, hybrid introgression, and codivergence in the Nasonia species complex. Evolution 63:165–183.
Minor comments
L12-13: Please reconsider the framing of this work (see major concerns above).
L31: “huge losses” to what? Please rephrase
L31-32: Please explain (shortly) the link between pest control and species identification. Why is identification important, and how can that help avoid losses?
L46: If the three species groups discussed here are the same as above, please add “the” before “three species groups”.
L48-50: Any references to the studies referred to here?
L53-56: These sentences seem a little out of place – seems odd to assume that morphological variation in male tibia is a polymorphism and not just continuous variation without knowing about Frolov’s work explained below, and so far you have not stated anything controversial. You could perhaps move these sentences much earlier in the introduction (starting from line 34) or later (starting from line 71).
L75-77: Check both contents and grammar. In reference number 15, authors suggest using nuclear markers rather than mitochondrial? Is it whole mitochondrial genomes or just mitochondrial genes or the barcode region that are used the most?
L78: What do you mean by “to date”? Your first reference here is nearly 30 years old and then you only have one newer reference, which concerns damselflies only?
L90-92: How many individuals of each species were sampled? If only one, it would be nice to know the sex of the sampled individual, although not often stated. Could be added to the Supplementary Table S1.
L92: In what temperature did you keep the dry samples?
L94-95: Nothing is mentioned about the DNA extraction method, PCR protocol or sequencing method. Please elaborate.
L96: If the samples were “up to 99.8% similar”, i.e. at best 99.8% similar, what was the worst match you considered as correct species identification?
L97: Thorax is a good choice for this purpose.
L103: Where was sequencing done?
L108-109: What is meant by “correctly annotated” here – do you mean that the annotations were corrected prior to illustration?
L112-113: I am not familiar with the method. Please explain what exactly do you manually edit using Adobe Illustrator CC2019?
L123 and 125: Use either ingroup and outgroup or in-group and out-group, but not mixed.
L134: Why did you use four datasets? This is also unclear in the results section.
L159-160: These values are not the same for all species in Table 2, and the values given here do not seem to be average values either. Please clarify.
L161-171: Was the information on intragenic spacers used for inferring the phylogenetic tree in this study, or is this merely descriptive?
L180-181: Grammar – “with a total length of about”
L189-190: This sentence is unclear to me – what does “are partial to ending with T or A” mean?
You talk a lot about codon usage – is that of importance in inferring phylogenetic relationships also? Were the ending codons included in the PCG sequences used for the phylogenetic analysis?
L210-211: Instead of “totally the same”, could use “of all tRNA’s, the trnG, trnS1(AGN) and trnT are identical among species”
L232-233: Involved in controlling?
L235: “motif ‘TTAGA’ followed by a long”?
L245-246: Slightly high is a bit vague. Perhaps rephrase the sentence, e.g. nad6 and atp6 were found to have slightly higher nucleotide diversity than the other genes ........ (, respectively), and atp8 shows the lowest value...
L251: How reliable is the estimation of synonymous substitutions in this dataset? How are the synonymous substitutions detected or simulated?
L254-255: Was this the same in all species? Any reason to think that this is an outlier? Also, weirdly, this same gene has the lowest nucleotide diversity...
L262-263: “due to it’s easy amplification”
L274: the three species groups are not monophyletic according to this tree - only the third one is, and the other two are paraphyletic.
L281: If the lengths of the branches are proportional to evolutionary changes, I would not state that O. penitalis is particularly closely related to the third group (quite long branch). Especially since the support values are not 100%.
L295: …additional sampling of taxa…
L297-298: Please elaborate!
L313-314: Yes, but you have not put your results in robust test against other methods.
L330: missing a comma
Figure 1: The font around the outermost circles on each mitogenome is too small.
Table 2: Horizontal lines between each set of species would help reading the table.
Figure 3: The font associated with arrows at the figure bottom middle are slightly too small.
Figure 4: Could you emphasise still where the 'TTAGA' block is, is it at the end of the first 25bp string highlighted? How do these colours look like in black and white, and are they visible for colour-blind? Otherwise, a nice figure, as the others also.
Figure 5A: Could make the legend for the Pi red line bigger.
Figure 5 legend, line 259: What genetic distance? On average?
Author Response
Response to Reviewer 1 Comments
Review of a manuscript titled “Complete mitogenomic structure and phylogenetic implications of the genus Ostrinia (Lepidoptera: Crambidae)”
This manuscript was a concise and clear representation -- with nice illustrations -- of a body or work revealing the complete mitogenomes of four moth species in the genus Ostrinia, some of which are notorious pests of important crop species (e.g. corn). I was not familiar with all the methods used for describing and illustrating mitogenomes, but considered the bioinformatics and results sufficiently and clearly described. I have no major concerns regarding the soundness of the work. However, given the sampling of both taxa (only four species from two subgroups) and molecular resources (limited to mitochondria only), I found the current framing of the work somewhat unsuited.
The authors justify their efforts by providing more resolution to the species -level phylogeny for these moths, many of which lack clear morphological apomorphies (traits unique to particular species), and also by providing molecular resources for various further studies on the selected species. To avoid potential pitfalls detailed below, I suggest rephrasing the study aims, and reframing introduction and discussion more towards the latter of the two incentives above. Although phylogenetic relationships within Ostrinia are interesting, I see more value in the molecular resources provided by this work than in the information value added to the already existing phylogenetic hypothesis.
Main concerns
1) The purpose of the study (from wide, societal point of view): At the beginning of the introduction (lines 29-34) the authors (referred to as “you” from now on) state that this group of species (Ostrinia) includes pests, are hard to identify and have a spurious phylogeny. However, I am failing to see the link between species identification and pest management – how are these species controlled? If a general pesticide is used, why is species identification necessary? If the species have distinct pheromones that are used in pest management, then please make this connection clear in the introduction. Furthermore, if knowing the full mitogenome is helpful for the management in some way, please explain how.
Response 1: Thank you for pointing this out. This section was revised to ‘Globally, Tichogramma spp. [7], transgenic crops [8], and sex pheromone traps [9] have been conducted as control strategies to reduce threat of crops caused by Ostrinia spp. Therefore, accurate species-level identification is very important for pests management as different species exhibit distinct responses to specific biocontrol agents and pesticides’. On line 31-35
[7] Wu, L.H.; Hill, M.P.; Thomson, L.J.; Hoffmann, A.A. Assessing the current and future biological control potential of Trichogramma ostriniae on its hosts Ostrinia furnacalis and Ostrinia nubilalis. Pest Manag. Sci. 2018, 74, 1513-1523, doi:10.1002/ps.4841.
[8] Bourguet, D.; Chaufaux, J.; Seguin, M.; Buisson, C.; Hinton, J.L.; Stodola, T.J.; Porter, P.; Cronholm, G.; Buschman, L.L.; Andow, D.A. Frequency of alleles conferring resistance to Bt maize in French and US corn belt populations of the European corn borer, Ostrinia nubilalis. Theor. Appl. Genet. 2003, 106, 1225-1233, doi:10.1007/s00122-002-1172-1.
[9] Gemeno, C.; Sans, A.; Lopez, C.; Albajes, R.; Eizaguirre, M. Pheromone antagonism in the European corn borer moth Ostrinia nubilalis. J. Chem. Ecol. 2006, 32, 1071-1084, doi:10.1007/s10886-006-9046-7.
2) The purpose of the study (from narrow, phylogenetic point of view): Study aims (on lines 82-84) are stated as follows: “…to sequence the complete mitogenomes of O. furnacalis, O. nubilalis, O. scapulalis, and O. zealis and reconstruct the phylogeny within the three species groups of the genus Ostrinia.” Please clarify, whether you refer to the three species groups as delineated by Mutuura and Monroe (1970), or to the three subgroups within Mutuura and Monroe’s group III. Based on sampled taxa, the aim was to identify species relationships among the latter smaller group, as only four species all belonging to this subgroup were sampled.
Response 2: We have carefully considered the comments. Indeed, there were six mitogenomic sequences used for our analyses, including four newly sequenced complete mitogenomes in species group III (O. furnacalis, O. nubilalis, O. scapulalis, and O. zealis ), one complete mitogenome in group II (O. palustralis), and one nearly complete mitogenomic sequences in group I (O. penitalis). The primary aim of this paper was to infer phylogenetic relationships within three species groups of the genus Ostrinia other than group III.
3) Not enough evidence to “debate” the existing phylogenetic hypothesis?: According to your introduction, Kim et al. (1999) and Yang et al. (2011) had already resolved the phylogeny of the Ostrinia species considered here, using also mitochondrial genes – see lines 67-69 and your results. You also state on line 287-289: “Our findings highly support the phylogenetic relationship within the third group revealed by mitochondrial COI and COII in previous studies [11,52]”. Therefore it seems, that you are not really “debating” the current hypothesis of the real phylogeny, but rather just providing more detailed picture of what can be inferred using mitochondrial data only. Even though you have higher support for species relationships using whole mitogenomes than Kim et al. (1999) and Yang et al. (2011) using either COII or COI, your results are essentially the same. This makes me question, why did you decide to test the phylogenetic hypothesis using mitochondrial markers only? Did you expect to find different results based on more mitochondrial genes or regions than with COII or COI alone? To me, this does not constitute as a robust test for the species relationships (see my points number 4-6 below).
If you want to make the novelty of your results clear, please clarify what is different between the previous results and your results. However, if the motivation of this study was rather to solve the mitochondrial genomes fully for some further use than to “debate the phylogenetic relationships of the genus Ostrinia”, as you state in the very first sentence of your abstract (line 12), please change the abstract and introduction accordingly. The last sentence of the introduction (lines 85-87) describes your motivation nicely, but could be expanded on. I.e. I am not fully convinced that resolving the mitogenome will help facilitate studies on “e.g. morphological character adaptations, host preferences, biogeographical origins or pheromones”, unless you provide some examples of how this can be done (i.e. references for examples of each of the suggested applications, or explanation on how this is relevant). Also, in the last paragraph of your discussion, you could tone down the sentences debating the phylogeny (i.e. line 289-290), where you propose that the division based on male mid-tibia needs to be reconsidered – this is already evident from previous studies.
Response 3: As stated above, we attempt to fully analyze the mitogenome characteristics and infer phylogenetic relationships based on mitogenomes of the genus Ostrinia. Our findings show that the relationships within this genus are (O. palustralis + (O. penitalis + (O. zealis + (O. furnacalis + (O. nubilalis + O. scapulalis))))), indicating that O. penitalis might not be the most primitive species as it may not constitute the basal elements of the genus instead of O. palustralis which is mainly different from Mutuura & Munroe’s classification. Kim et al. (1999), Yang et al. (2011) were not able to conclude the similar results as they did not include O. penitalis in their studies. The sentence (line 289-290) you mentioned was rephrased to ‘In other words, our results confirmed that the division of three subgroups within the third species based on male mid-tibia by Mutuura & Munroe (1970) needs to be reconsidered.’ On line 302-304
4) Incongruent mitochondrial vs. nuclear gene trees: If the main aim of this study was to resolve the true phylogenetic relationships, why did you choose to use the same genetic data source already used, and not e.g. nuclear genes or more variable microsatellites? Trees based on mitochondrial genes have often been found to deviate from trees inferred from nuclear genes, see e.g. Funk and Omland 2003, Leache and McGuire 2006. Mitochondria may be introgressed between closely related species if they interbreed, which would result in a different phylogeny inferred from mitochondrial genes than from nuclear genes. Have you checked the possibility for interbreeding in Ostrinia?
Funk, D. J. & Omland, K. E. Species-level paraphyly and polyphyly: frequency, causes, and consequences, with insights from animal mitochondrial DNA. Annu. Rev. Ecol. Evol. Syst. 34, 397–423 (2003).
Leache, A. D. & McGuire, J. A. Phylogenetic relationships of horned lizards (Phrynosoma) based on nuclear and mitochondrial data: evidence for a misleading mitochondrial gene tree. Mol. Phylogenet. Evol. 39, 628–644 (2006).
Response 4: We do agree sometimes the phylogenetic tree inferred by mitochondrial genes conflicts with those by nuclear genes. However, many previous studies show that mt phylogenomics are highly congruent with nuclear gene analyses for majority of groups in Lepidoptera [1]. Within the Ostrinia, the phylogenetic relationships based on both the mitochondrial genes or nuclear genes have consistent results in previous studies [2-4].
[1]Cameron, S.L. Insect Mitochondrial Genomics: Implications for Evolution and Phylogeny. Annu. Rev. Entomol. 2014, 59, 95-117, doi:10.1146/annurev-ento-011613-162007.
[2] Lassance, J.M.; Lienard, M.A.; Antony, B.; Qian, S.; Fujii, T.; Tabata, J.; Ishikawa, Y.; Lofstedt, C. Functional consequences of sequence variation in the pheromone biosynthetic gene pgFAR for Ostrinia moths. Proc. Natl. Acad. Sci., India, Sect. A 2013, 110, 3967-3972, doi:10.1073/pnas.1208706110.
[3] Kim, C.G.; Hoshizaki, S.; Huang, Y.P.; Tatsuki, S.; Ishikawa, Y. Usefulness of mitochondrial COII gene sequences in examining phylogenetic relationships in the Asian corn borer, Ostrinia furnacalis, and allied species (Lepidoptera: Pyralidae). Appl. Entomol. Zool. 1999, 34, 405-412, doi: 10.1303/aez.34.405.
[4] Yang, R.S.; Wang, Z.Y.; He, K.L. Genetic diversity and phylogeny of the genus Ostrinia (Lepidoptera: Crambidae) inhabiting China inferred from mitochondrial COI gene. J. Nanjing Agric. Univ. 2011, 34(5), 73—80.
5) Incomplete lineage sorting: If introgression occurs only in some populations but not others, it may result in incomplete lineage sorting. Now if you sampled only from one population, and one individual of each species, how can you account for possible incomplete lineage sorting? This effect has been shown to be a likely cause for incongruent species trees between mtDNA and nuclear DNA (e.g. Wang et al. 2018).
Wang, K., Lenstra, J. A., Liu, L., Hu, Q., Ma, T., Qiu, Q., & Liu, J. (2018). Incomplete lineage sorting rather than hybridization explains the inconsistent phylogeny of the wisent. Communications biology, 1(1), 1-9.
Response 5: We do agree that it is generally risky to infer the species trees if introgression occurs. However, the validity of none of the species is justified by mitochondrial sequences only, but that all species show diagnostic morphological differences too in our study. In addition, our results show that the phylogenetic relationships among species based on mitogenomes was highly congruent with the topology inferred by nuclear DNA [1].
[1] Lassance, J.M.; Lienard, M.A.; Antony, B.; Qian, S.; Fujii, T.; Tabata, J.; Ishikawa, Y.; Lofstedt, C. Functional consequences of sequence variation in the pheromone biosynthetic gene pgFAR for Ostrinia moths. Proc. Natl. Acad. Sci., India, Sect. A 2013, 110, 3967-3972, doi:10.1073/pnas.1208706110.
6) Limited sampling effort: If your aim was to resolve species relationships within the three subgroups in the third species group in Ostrinia, then this sampling is fine. However, if you aimed for the three species groups, why did you not sample more species? If you aimed to provide resources for further studies on e.g. pheromones, why not sequence genes associated with pheromones or even whole genomes for these species? If the reason was cost-efficiency, please state so.
Response 6: The primary aim of this paper was to infer phylogenetic relationships within three species groups of the genus Ostrinia based on mitochondrial genome data due to its cost-efficiency. Because of the limit samples, we only included four representative species of the third species group. However, as your suggestion additional samples for remaining species would be helpful for reveal the deep relationships within the third species.
7) Wolbachia infection: Did you check whether any of you samples were infected with Wolbachia, or detect any traces of a possible Wolbachia infection, which may have caused changes in the mitogenome of any of the species? Wolbachia infection may cause a selective sweep in the genome reducing sequence diversity, but also increased levels of DNA introgression (perhaps also between species). For examples of genomic effects of Wolbachia in Lepidoptera, see e.g. Jiggins 2003, Narita et al. 2006, Narita et al. 2007 and for introgression also Raychoudhury et al. 2009 on wasps. Please elaborate whether you checked for Wolbachia infection, and whether you find it necessary to consider the possibility and importance of Wolbachia infection to the structure of these four new mitogenomes and the inferred phylogeny?
Jiggins, F. M. 2003. Male‐killing Wolbachia and mitochondrial DNA: selective sweeps, hybrid introgression and parasite population dynamics. Genetics 164:5–12.
Narita, S., M. Nomura, Y. Kato, and T. Fukatsu. 2006. Genetic structure of sibling butterfly species affected by Wolbachia infection sweep: evolutionary and biogeographical implications. Mol. Ecol. 15:1095–1108.
Narita, S., M. Nomura, Y. Kato, O. Yata, and D. Kageyama. 2007. Molecular phylogeography of two sibling species of Eurema butterflies. Genetica 131:241–253.
Raychoudhury, R., L. Baldo, D. C. S. G. Oliveira, and J. H. Werren. 2009. Modes of acquisition of Wolbachia: horizontal transfer, hybrid introgression, and codivergence in the Nasonia species complex. Evolution 63:165–183.
Response 7: We considered, but ruled out, the possibility that might derive from a symbiont infection such as Wolbachia. Although the difference in mtDNA could be caused by infection with different strains of Wolbachia, but Wolbachia infection does not always show up during sequencing and tends to promote mitochondrial introgression rather than divergence (Braig et al., 1998, Hurst & Jiggins 2005, Frézal & Leblois, 2008, Smith & Fisher 2009, Muñoz et al., 2011). In addition, we used both sequences and morphological characters to delimit species differences.
Minor comments
L12-13: Please reconsider the framing of this work (see major concerns above).
Response: We revised it following your suggestion. The sentence was revised to ‘To understand mitogenome characteristics and reveal phylogenetic relationships of the genus Ostrinia, including several notorious pests of great importance for crops, we sequenced the complete mitogenomes of four species: Ostrinia furnacalis (Guenée, 1854), Ostrinia nubilalis (Hübner, 1796), Ostrinia scapulalis (Walker, 1859) and Ostrinia zealis (Guenée, 1854).’ On line 12-13
L31: “huge losses” to what? Please rephrase
Response: The sentence was revised to ‘…that cause huge losses of crops around the world.’ On line 31
L31-32: Please explain (shortly) the link between pest control and species identification. Why is identification important, and how can that help avoid losses?
Response: We added an explanation about the link between pest control and species identification. ‘Globally, Tichogramma spp. [7], transgenic crops [8], and sex pheromone traps [9] have been conducted as control strategies to reduce threat of crops caused by Ostrinia spp. Therefore, accurate species-level identification is very important for pests management as different species exhibit distinct responses to specific biocontrol agents and pesticides.’ On line 31-35
L46: If the three species groups discussed here are the same as above, please add “the” before “three species groups”.
Response: It is the same “three species groups” and we add ‘the’ before ‘three species groups’. On line 50
L48-50: Any references to the studies referred to here?
Response: We add a reference to support this viewpoint. Mutuura, A.; Munroe, E. Taxonomy and distribution of the European corn borer and allied species: genus Ostrinia (Lepidoptera: Pyralidae). Mem. Entomol. Soc. Can. 1970, 102, 1-112, doi:10.4039/entm10271fv. On line 52
L53-56: These sentences seem a little out of place – seems odd to assume that morphological variation in male tibia is a polymorphism and not just continuous variation without knowing about Frolov’s work explained below, and so far you have not stated anything controversial. You could perhaps move these sentences much earlier in the introduction (starting from line 34) or later (starting from line 71).
Response: Thanks for your suggestion and we moved these sentences to the end of next paragraph. On line 74-77
L75-77: Check both contents and grammar. In reference number 15, authors suggest using nuclear markers rather than mitochondrial? Is it whole mitochondrial genomes or just mitochondrial genes or the barcode region that are used the most?
Response: We rechecked this section and rephrased it to ‘The mitochondrial genome was widely used to study taxonomy, population genetics and phylogenetic relationships because of the extremely low rate of recombination, maternal inheritance, and faster evolutionary rate compared to nuclear DNA [19-22]. The mitogenome in general provides more phylogenetically informative due to significant sequence differences could be observed among related species compared to a single mitochondrial gene or few genes [23-27]’. On line 81-86
L78: What do you mean by “to date”? Your first reference here is nearly 30 years old and then you only have one newer reference, which concerns damselflies only?
Response: We have made the changes accordingly based on the comments. On line 81-86
L90-92: How many individuals of each species were sampled? If only one, it would be nice to know the sex of the sampled individual, although not often stated. Could be added to the Supplementary Table S1.
Response: Only one adult male of each species was used for DNA extraction in this study. The sex of sample was added into Supplementary Table S1 as you suggested. On line 330
L92: In what temperature did you keep the dry samples?
Response: We kept the dry samples under 20℃ condition. On line 97
L94-95: Nothing is mentioned about the DNA extraction method, PCR protocol or sequencing method. Please elaborate.
Response: The details of DNA extraction was added in the text. ‘In order to confirm the identities, we generated the 658 base pair (bp) barcode region of COI sequences from a single leg using DNAeasy DNA Extraction kit following the manufacturer’s protocols. LepF1/LepR1 primers were used to PCR amplify DNA fragments from COI gene [31]. DNA products were subsequently bidirectionally sequenced by Sangon Biotechnology Co. ltd (Shanghai, China). The COI sequences were cross-checked against the Barcode of Life Database [32]’ On line 99-103
[31] Hebert, P.D.N.; Penton, E.H.; Burns, J.M.; Janzen, D.H.; Hallwachs, W. Ten species in one: DNA barcoding reveals cryptic species in the neotropical skipper butterfly Astraptes fulgerator. Proc. Natl. Acad. Sci., India, Sect. A 2004, 101, 14812-14817, doi:10.1073/pnas.0406166101.
L96: If the samples were “up to 99.8% similar”, i.e. at best 99.8% similar, what was the worst match you considered as correct species identification?
Response: For species identification of Ostrinia, less 99.0% similarity was the worst match.
L97: Thorax is a good choice for this purpose.
Response: Thorax can provide more tissue and muscle for DNA extraction.
L103: Where was sequencing done?
Response: Complete mitogenomes were sequenced at the Biomarker Technologies Co. ltd (Beijing, China). On line 112
L108-109: What is meant by “correctly annotated” here – do you mean that the annotations were corrected prior to illustration?
Response: To illustrate mitogenomic circular maps, we carefully double-checked the mitogenome annotation based on published Ostrinia mitogenomes.
L112-113: I am not familiar with the method. Please explain what exactly do you manually edit using Adobe Illustrator CC2019?
Response: The tRNAs secondary structure was predicted from the MITOS Web Serve (http://mitos2.bioinf.uni-leipzig.de/index.py) and original figures were generated. All the figures were subsequently modified and edited using Adobe Illustrator CC2019.
L123 and 125: Use either ingroup and outgroup or in-group and out-group, but not mixed.
Response: We followed your suggestion to use ‘ingroup’ and ‘outgroup’ in the ms. On line 132-134
L134: Why did you use four datasets? This is also unclear in the results section.
Response: Based on previous studies, 13PCGs and two RNAs were widely used to construct phylogenetic relationship. Moreover, replicate analyses including or excluding the third codons could test variability in the phylogenetic performance [20, 44]. On line 142-144
[20] Cameron, S.L. Insect Mitochondrial Genomics: Implications for Evolution and Phylogeny. Annu. Rev. Entomol. 2014, 59, 95-117, doi:10.1146/annurev-ento-011613-162007.
[44] Kim, M.J.; Kang, A.R.; Jeong, H.C.; Kim, K.G.; Kim, I. Reconstructing intraordinal relationships in Lepidoptera using mitochondrial genome data with the description of two newly sequenced lycaenids, Spindasis takanonis and Protantigius superans (Lepidoptera: Lycaenidae). Mol. Phylogenet. Evol. 2011, 61, 436-445, doi:10.1016/j.ympev.2011.07.013
L159-160: These values are not the same for all species in Table 2, and the values given here do not seem to be average values either. Please clarify.
Response: The average values for all species were added in the text accordingly. On line 168-172
L161-171: Was the information on intragenic spacers used for inferring the phylogenetic tree in this study, or is this merely descriptive?
Response: It was a description of mitogenome organization in this part.
L180-181: Grammar – “with a total length of about”
Response: The sentence was modified to ‘All PCGs among the five complete mitogenomes are similar in general character with a total length about 11,163 bp (Table S3).’ On line 192-193
L189-190: This sentence is unclear to me – what does “are partial to ending with T or A” mean?
You talk a lot about codon usage – is that of importance in inferring phylogenetic relationships also? Were the ending codons included in the PCG sequences used for the phylogenetic analysis?
Response: This sentence was modified to ‘Overall codon usage analysis indicated that the codon ending up with T or A is more frequent than C or G.’ This section shows the relative synonymous codon usage which might not be relevant with inferring phylogenetic relationships. The stop codons were removed from the PCG sequences used for the phylogenetic analysis. On line 202-203
L210-211: Instead of “totally the same”, could use “of all tRNA’s, the trnG, trnS1(AGN) and trnT are identical among species”
Response: We revised this sentence based on your suggestion. On line 222-223
L232-233: Involved in controlling?
Response: We revised the sentence. On line 244-246
L235: “motif ‘TTAGA’ followed by a long”?
Response: We rephrased the sentence. On line 248-249
L245-246: Slightly high is a bit vague. Perhaps rephrase the sentence, e.g. nad6 and atp6 were found to have slightly higher nucleotide diversity than the other genes ........ (, respectively), and atp8 shows the lowest value...
Response: This sentence was revised based on your suggestion. On line 257-258
L251: How reliable is the estimation of synonymous substitutions in this dataset? How are the synonymous substitutions detected or simulated?
Response: In order to rechecked the synonymous substitution, the evolution rate analysis among 13PCGs was conducted using KaKs_Calculator again. The results indicated that all PCGs values are reliable (P-Value< 0.01) excepted for atp8 (P-Value =0.396). In this study the synonymous substitutions were detected based on Jukes-Cantor (JC) models which assumes that all substitutions have equal rates and equal nucleotide frequencies.
Sequence |
Ka |
Ks |
Ka/Ks |
P-Value (Fisher) |
atp6 |
0.0204649 |
0.201065 |
0.101782 |
1.87994e-029 |
atp8 |
0.0196233 |
0.0319376 |
0.614426 |
0.396887 |
cox1 |
0.0019702 |
0.218901 |
0.00900041 |
9.95574e-123 |
cox2 |
0.00583329 |
0.194917 |
0.029927 |
3.48304e-041 |
cox3 |
0.0125928 |
0.174421 |
0.0721976 |
7.18538e-035 |
cytb |
0.0111075 |
0.215378 |
0.0515724 |
8.5353e-066 |
nad1 |
0.00990573 |
0.213151 |
0.0464729 |
3.19619e-056 |
nad2 |
0.0112562 |
0.146703 |
0.0767278 |
1.26519e-029 |
nad3 |
0.00347421 |
0.186782 |
0.0186004 |
2.44204e-021 |
nad4 |
0.0114995 |
0.192458 |
0.0597506 |
3.03169e-064 |
nad4l |
0.0257399 |
0.120556 |
0.213509 |
1.15154e-005 |
nad5 |
0.0100524 |
0.136004 |
0.0739131 |
8.1836e-057 |
nad6 |
0.0191126 |
0.229699 |
0.0832074 |
1.80106e-027 |
[1] Zhang, Z.; Li, J.; Zhao, X.-Q.; Wang, J.; Wong, G.K.-S.; Yu, J.; bioinformatics. KaKs_Calculator: calculating Ka and Ks through model selection and model averaging. Genomics, proteomics 2006, 4, 259-263, doi:10.1016/S1672-0229(07)60007-2
[2] Jukes, T.; Cantor, C. Evolution of protein molecules. In ‘‘Mammalian Protein Metabolism’’(HN Munro, Ed.), pp. Pages21–132. Academic Press, New York, NY: 1969
L254-255: Was this the same in all species? Any reason to think that this is an outlier? Also, weirdly, this same gene has the lowest nucleotide diversity...
Response: Because of the lowest nucleotide diversity and extremely highest values we inferred that the atp8 may be an outlier, similar results were observed in previous study [62]. On line 266-267
[62] Ma, L.; Liu, F.; Chiba, H.; Yuan, X. The mitochondrial genomes of three skippers: Insights into the evolution of the family Hesperiidae (Lepidoptera). Genomics 2019, 112, 432-441, doi:10.1016/j.ygeno.2019.03.006.
L262-263: “due to it’s easy amplification”
Response: We rephrased the sentence based on your suggestion. On line 275
L274: the three species groups are not monophyletic according to this tree - only the third one is, and the other two are paraphyletic.
Response: This sentence was rephrased to ‘Both ML and BI trees revealed that the third species group is monophyletic.’ On line 287
L281: If the lengths of the branches are proportional to evolutionary changes, I would not state that O. penitalis is particularly closely related to the third group (quite long branch). Especially since the support values are not 100%.
Response: Thanks for your suggestion, we rephrased this sentence to ‘The representative of the first species group O. penitalis might not be the most primitive species as it may not constitute the basal elements of the genus instead of O. palustralis, which is mainly different from Mutuura & Munroe’s classification.’ On line 294-296
L295: …additional sampling of taxa…
Response: This sentence was revised based on your suggestion. On line 309-311
L297-298: Please elaborate!
Response: We removed this sentence.
L313-314: Yes, but you have not put your results in robust test against other methods.
Response: Thanks for your professional suggestion and we removed this sentence.
L330: missing a comma
Response: We added a comma. On line 342
Figure 1: The font around the outermost circles on each mitogenome is too small.
Response: Thanks for your suggestion and we adjusted the font to be clear. On line 184
Table 2: Horizontal lines between each set of species would help reading the table.
Response: The horizontal lines were added into the table2. On line 190
Figure 3: The font associated with arrows at the figure bottom middle are slightly too small.
Response: The font was adjusted. On line 226
Figure 4: Could you emphasise still where the 'TTAGA' block is, is it at the end of the first 25bp string highlighted? How do these colours look like in black and white, and are they visible for colour-blind? Otherwise, a nice figure, as the others also.
Response: The ‘TTAGA’ is at the end of the first block (length of 25bp) and we regenerated this figure following your suggestion. On line 250
Figure 5A: Could make the legend for the Pi red line bigger.
Response: We have enlarged this legend following your kindly suggestion. On line 268
Figure 5 legend, line 259: What genetic distance? On average?
Response: The average genetic distances are shown in the chart. On line 268

Reviewer 2 Report
In their manuscript "Complete mitogenomic structure and phylogenetic implications of the genus Ostrinia (Lepidoptera: Crambidae)" Zhou et al. discuss the phylogenetic relationship of the important genus Ostrinia -which includes several notorious crop pests- by sequencing the mitogenome of four species of this genus. They further compare these mitogenomes.
In their introduction, authors give a good overview of the systematics based on morphological characters and explain how morphology-based taxonomy contradicts with molecular data and point out that there is only one complete mitogenome of genus Ostrinia.
Material and methods are of good scientific soundness.
In the results and discussion, authors compare the MITOgenomic structure and organization, nucleotide diversity, etc. I think these paragraphs are original and novel.
The aim state by the authors is to reconstruct "the phylogeny within the three species groups of the genus Ostrinia with the four species they sequenced." Furthermore, authors propose several taxonomic implications: 1."that O. penitalis might not be the most primitive species", 2. "the division of the third species group based on male mid-tibia needs to be reconsidered", 3. "all our results indicate that the mitogenomes provide robust evidence to clarify the phylogenetic relationships within the genus Ostrinia". I think based only on the mitogenome sequence and given the small samples size, authors should be more careful with their taxonomic implications. Moreover, to me it is not clear how the present study "will facilitate future systematic and evolutionary studies of the genus Ostrinia (e.g. morphological character adaptions, host preferences, biogeographic origins, and pheromones), and
promote studies on the biological control of this economically significant group". I suggest to adjust hte text. It is a nice manuscript, so there is no need in trying to oversell it.
Line 146: Misleading: "3.1 Genome structure and organization": Authors are not comparing genomes but mitogenomes!
Throughout the text: Use of ‘phylogeny' should be avoided when reporting the results of an analysis (there is only one true phylogeny). Use ‘phylogenetic analysis', ‘phylogenetic tree' or similar.
Author Response
Response to Reviewer 2 Comments
In their manuscript "Complete mitogenomic structure and phylogenetic implications of the genus Ostrinia (Lepidoptera: Crambidae)" Zhou et al. discuss the phylogenetic relationship of the important genus Ostrinia -which includes several notorious crop pests- by sequencing the mitogenome of four species of this genus. They further compare these mitogenomes.
In their introduction, authors give a good overview of the systematics based on morphological characters and explain how morphology-based taxonomy contradicts with molecular data and point out that there is only one complete mitogenome of genus Ostrinia.
Material and methods are of good scientific soundness.
In the results and discussion, authors compare the mitogenomic structure and organization, nucleotide diversity, etc. I think these paragraphs are original and novel.
The aim state by the authors is to reconstruct "the phylogeny within the three species groups of the genus Ostrinia with the four species they sequenced." Furthermore, authors propose several taxonomic implications: 1."that O. penitalis might not be the most primitive species", 2. "the division of the third species group based on male mid-tibia needs to be reconsidered", 3. "all our results indicate that the mitogenomes provide robust evidence to clarify the phylogenetic relationships within the genus Ostrinia".
I think based only on the mitogenome sequence and given the small samples size, authors should be more careful with their taxonomic implications. Moreover, to me it is not clear how the present study "will facilitate future systematic and evolutionary studies of the genus Ostrinia (e.g. morphological character adaptions, host preferences, biogeographic origins, and pheromones), and promote studies on the biological control of this economically significant group". I suggest to adjust the text. It is a nice manuscript, so there is no need in trying to oversell it.
Response: Thanks for your suggestion and we removed the sentence that might cause confusion. We reorganized the purpose to make it clear and readable.
There were six mitogenomic sequences used for our analyses, including four newly sequenced complete mitogenomes in species group III (O. furnacalis, O. nubilalis, O. scapulalis, and O. zealis ), one complete mitogenome in group II (O. palustralis), and one nearly complete mitogenomic sequences in group I (O. penitalis). The primary aim of this paper was to infer phylogenetic relationships within three species groups of the genus Ostrinia
Line 146: Misleading: "3.1 Genome structure and organization": Authors are not comparing genomes but mitogenomes!
Response: Thanks for your professional suggestion we revised this title to ‘Mitogenome structure and organization’. On line 158
Throughout the text: Use of ‘phylogeny' should be avoided when reporting the results of an analysis (there is only one true phylogeny). Use ‘phylogenetic analysis', ‘phylogenetic tree' or similar.
Response: The suggested changes were made throughout the text.

Round 2
Reviewer 1 Report
Comments for authors, revision round II
I find the corrections made by the authors sufficient, and I hope the authors share my view on that the already high quality of the manuscript was further improved in accuracy during this revision process. In particular, I am pleased with the added references, showing that the authors have considered all relevant aspects in their study.
I have few further suggestions for polishing the language of few modified sentences as specified below:
Line 33-35: Modify “Therefore, accurate species-level identification is very important for pests management as different species exhibit distinct responses to specific biocontrol agents and pesticides.” into
“As different species exhibit distinct responses to specific biocontrol agents and pesticides, accurate species-level identification is very important in pest management.”
Line 80: “was widely used” into “has been widely used”
Line 83: “provides more phylogenetically informative due to” into “provides more phylogenetic information due to”
Line 265-266: This may have to be split into two sentences or explained more, but I have no good suggestion. Perhaps wait for the editing process.
Line 294-295: “…, which is mainly different from Mutuura & Munroe’s classification.” into “…, which is the main difference to Mutuura & Munroe’s (1970) classification.”
Author Response
Response to Reviewer 1 Comments
I find the corrections made by the authors sufficient, and I hope the authors share my view on that the already high quality of the manuscript was further improved in accuracy during this revision process. In particular, I am pleased with the added references, showing that the authors have considered all relevant aspects in their study.
I have few further suggestions for polishing the language of few modified sentences as specified below:
Line 33-35: Modify “Therefore, accurate species-level identification is very important for pests management as different species exhibit distinct responses to specific biocontrol agents and pesticides.” into
“As different species exhibit distinct responses to specific biocontrol agents and pesticides, accurate species-level identification is very important in pest management.”
Response: Thanks for your suggestion, the sentence was revised. On line 33-34
Line 80: “was widely used” into “has been widely used”
Response: This sentence was revised based on your suggestion. On line 81
Line 83: “provides more phylogenetically informative due to” into “provides more phylogenetic information due to”
Response: We revised this sentence based on your suggestion. On line 84
Line 265-266: This may have to be split into two sentences or explained more, but I have no good suggestion. Perhaps wait for the editing process.
Response: Thanks for your suggestion, we rephrased this sentence to ‘In addition, cox1 exhibits the strongest purifying selection with the lowest Ka/Ks value (0.009). Both nad6 (0.083) and atp6 (0.102) show slightly higher Ka/Ks values than most of other PCGs, indicating they are likely to be under a relaxed purifying selection.’ On line 267-269
Line 294-295: “…, which is mainly different from Mutuura & Munroe’s classification.” into “…, which is the main difference to Mutuura & Munroe’s (1970) classification.”
Response: We revised this sentence based on your suggestion. On line 299-300
